# Evolution of Microstructure and Texture in Grain-Oriented 6.5% Si Steel Processed by Rolling with Intrinsic Inhibitors and Additional Inhibitors

**DOI:** 10.3390/ma16206731

**Published:** 2023-10-17

**Authors:** Ruiyang Liang, Chengqian Sun, Qingchun Li

**Affiliations:** 1School of Materials Science and Engineering, Liaoning University of Technology, Jinzhou 121000, China; 13029333119@163.com; 2State Key Laboratory of Metal Material for Marine Equipment and Application, Anshan 114000, China; 3Iron and Steel Research Institute of Ansteel Group Corporation, Anshan 114000, China

**Keywords:** 6.5% Si steels, magnetic properties, Goss texture, high-energy grain boundaries

## Abstract

A grain-oriented steel containing 6.5% Si, characterized by a notable Goss texture, was effectively manufactured through the rolling technique, incorporating both intrinsic inhibitors and additional inhibitors. This investigation focuses on tracking the development of texture and magnetic properties during the manufacturing process and delineates the mechanism underlying secondary recrystallization. The empirical findings clearly demonstrated the significant influence of nitriding duration and quantity on the secondary recrystallization process. In instances where additional nitrogen is absent, the intrinsic inhibitors alone do not lead to secondary recrystallization. However, when the nitriding duration is 90 s and the nitriding amount is 185 ppm, a complete secondary recrystallization structure with a strong Goss texture enables the finished products have excellent magnetic properties. The preferential growth of Goss grains is mainly governed by the enhanced mobility of high-energy (HE) grain boundaries. With the increase in annealing temperature, the occurrence of 20°–45° HE grain boundaries with Goss grains becomes more progressively frequent. At the secondary recrystallization temperature of 1000 °C, the frequency of 20°–45° HE grain boundaries with Goss grains reaches 62.7%, providing favorable conditions for the abnormal growth of Goss grains. This results in a secondary recrystallization structure predominantly characterized by a strong Goss texture. In light of these observations, the present study provides fundamental theoretical insights and serves as a valuable procedural guideline for the industrial manufacturing of 6.5% Si grain-oriented electrical steels.

## 1. Introduction

Fe-6.5 wt% Si alloys are ideal core materials for motors, transformers, and generators with excellent soft magnetic properties such as high permeability and saturation magnetization, almost zero magnetostriction, and low eddy current and hysteresis losses especially at high frequencies [1,2]. Forming and texture control are the key factors to obtaining high-quality high-silicon steel. Although the high-silicon steel strip rolling technique based on the plasticization and toughing process has made much progress [3,4,5,6], its texture control theory and magnetic properties remain to be significantly improved. Developing beneficial textures, such as Goss and cube textures, is a cost-effective way to achieve high magnetic properties for high-silicon steel. Goss ({110}<001>) orientation, due to the crystal orientation direction parallel to the rolling direction (RD), has better susceptibility to magnetization under directional magnetic fields and therefore, it mostly exists in grain-oriented silicon steel. Cube ({100}<001>) orientation, which has two <001> parallel to the rolling direction (RD) and transverse direction (TD), respectively, is a favorable orientation for both grain-oriented and non-oriented silicon steel. However, cube orientation is difficult to control in grain-oriented silicon steel and can easily rotate to other orientations during deformation [7,8].

So far the studies about high-silicon steel have been mainly focused on non-oriented high-silicon steel [9,10,11,12,13,14], and there are few reports about producing grain-oriented 6.5 wt% silicon steel by a rolling process; this is because the preparation process for grain-oriented silicon steel is complex, and the size, distribution, and quantity of inhibitor need to be accurately controlled, especially the texture control which requires an extreme level. In addition, increasing the silicon content can delay or hinder the development of secondary recrystallization, and stronger inhibitors are needed to suppress the grain growth, which undoubtedly increases the difficulty of preparing grain-oriented silicon steel. Patents concerning grain-oriented high-silicon steel for electric purposes were already granted in Japan during the 1990s. However, the widespread industrial utilization of this material has been limited due to the low production efficiency and specific technical challenges. The availability of theoretical studies on grain-oriented high-silicon steel has also been limited. In recent years, there have been pertinent literature publications concerning the preparation process of grain-oriented high-silicon steel [15,16,17]. These reports have involved an analysis and comparison of the magnetic properties of this steel with non-oriented high-silicon steel, as well as non-oriented and grain-oriented silicon steel containing 3% Si with the same sheet thickness. Our laboratory study also successfully prepared grain-oriented high-silicon steel by adding inhibitors in the early stage [18]. Nonetheless, the process stability is insufficient, and there is still significant room for improvement in the magnetic properties. In addition, the specific characteristics and behaviors of the secondary recrystallization are not well known, and should be further investigated.

Given the substantial practical and theoretical importance associated with investigating the manufacturing process and attributes of secondary recrystallization in 6.5% Si grain-oriented electrical steels, this paper explores a novel method for producing such steel. The proposed procedure entails the utilization of both intrinsic inhibitors at the outset and subsequent supplementary inhibitors in the production of 6.5 wt% Si grain-oriented electrical steels. 

## 2. Experimental Procedures

For this experiment, the standard approach using Cu-containing 3% Si grain-oriented silicon steel was employed, with certain adjustments made to the Al content. The specific composition of the material is as follows (in mass fraction, %): C 0.01, Si 6.5, Mn 0.12, S 0.008, Al 0.02, N 0.003, and Cu 0.2. Initially, the sample is cast into a 30 mm thin slab in a vacuum induction furnace. Subsequently, it undergoes five passes of hot rolling, commencing at 1150 °C and ending at 880 °C, resulting in a thickness reduction of 1.8 mm. Next, the sample is warm rolled at 450 °C, yielding a final sheet thickness of 0.25 mm. Following a 4 min temperature hold and a decarbonization annealing process at 850 °C, the oxidization layer is attenuated through mechanical polishing. A nitriding step occurs next in an atmosphere containing 15% NH_3_ and 75% H_2_ at 750 °C for 60–120 s. The sample is then coated with MgO and prepared for high-temperature annealing, as illustrated in Figure 1. The process begins in a nitrogen atmosphere, with the temperature rapidly increasing at 200 °C/h until it reaches 400 °C. A H_2_:N_2_ ratio of 1:1 is maintained as the temperature gradually increases up to 600 °C. At this point, the temperature is held for 4 h before rapidly increasing at a rate of 15 °C/h, reaching 1200 °C. Subsequently, the temperature is maintained for 5 h in a pure hydrogen atmosphere until the finished sheet is obtained. The “interruption method” was employed to study variations in sample structures, textures, and precipitation behaviors of second-phase particles during the high-temperature annealing process. In this interruption test, a sample subjected to 90 s nitriding was utilized. Starting from 900 °C, the sample was periodically taken out of the furnace at 50 °C intervals, allowing for the examination of changes in the sample structure and texture. The preparation process of 6.5 wt% Si grain-oriented electrical steels is illustrated in Figure 2. 

In this investigation, field-emission secondary electron microscopy (FESEM) was employed to observe the precipitates, and the compositions of the precipitated phases were analyzed using energy-dispersive X-ray spectroscopy (EDS). The texture of the samples was quantified and assessed with the aid of an Oxford Instruments HKL-Channel 5 EBSD system. The magnetic properties were evaluated using an electrical steel tester (MPG200D). 

## 3. Results

### 3.1. Evolution of Organization and Texture

Figure 3 displays the electron backscatter diffraction (EBSD) orientation maps depicting the characteristics of hot-rolled sheets in the lateral section. It is evident that the hot-rolled plate containing 6.5% Si content displayed a noticeable gradient in both microstructure and texture, closely resembling the characteristic microstructural features commonly observed in conventionally grain-oriented silicon steel. The surface and subsurface layers primarily consisted of dynamically recrystallized structures, with the predominant texture types being Goss, {112}<111>, and {110}<112>, as shown in Figure 3a–c,e. The Goss texture had fewer components compared to the {110}<112> texture components, which is related to the low-temperature hot rolling process. The center layer, experiencing solely plane stress, exhibited a deformed elongated structure characterized by the presence of {100}<021>, {113}<361>, and {111}<112> grains, as shown in Figure 3b,d. 

The Goss-oriented grains generated during hot rolling served as the origin of the Goss texture during secondary recrystallization annealing. Furthermore, the hot rolling process also generated {110}<112> and {112}<111> textures. These three textures are recognized as typical shear textures in grain-oriented silicon steel. It is important to highlight that the laboratory rolling conditions, in contrast to the hydraulic transmission employed in industrial settings, resulted in a rapid temperature decrease during the rolling process, intensifying the shearing impact of hot rolling. As a result, hot-rolled silicon steel subjected to laboratory rolling conditions often displays a higher prevalence of {110}<112> grains.

Figure 4a–c displays the EBSD orientation maps of warm-rolled sheets in the lateral section, revealing the prominent 20–45° shear bands and fragmented grains, as shown in Figure 4a,b. The warm-rolled sheet is dominated by γ textures and α textures, with {112}<110> textures showing the highest orientation density, as shown in Figure 4c. Furthermore, there were a few weakly present Goss orientation grains, as shown by the red color in Figure 4b. The limited quantity of Goss grains was mostly distributed within the fragmented grain area of the {111}<112> orientation in the subsurface layer. According to high-energy boundary theory, {111}<112> grains with significant deformation energy storage are favorable for the coalescence and preferential growth of Goss grains. However, in this particular experiment, the strength of the {111}<112> texture was noticeably inferior to that observed in traditional 3% Si grain-oriented silicon steel. This discrepancy is related to the reduction in deformation energy storage caused by the warm rolling process.

Next, Figure 5a–c depicts the EBSD orientation maps of the decarburized sample. The main textures observed in the decarburized plate were {100}<011>, {111}<112>, and {113}<361>, as shown in Figure 5c. The orientation imaging map reveals a uniform microstructure in the decarburized plate, with an average grain size of 16.43 µm. Notably, the central layer exhibited a larger grain size ranging from 30 to 40 µm, accompanied by a dominant {100}<021> texture, as shown in Figure 5a,b.

Figure 6a–f displays the EBSD orientation maps obtained at various nitriding durations, while Table 1 presents the statistical data regarding the nitriding quantity and average grain size corresponding to the different nitriding durations. It is apparent that the grain size remained unchanged as the nitriding time increased, with an average grain size ranging from 17 to 22 µm. This observation is ascribed to the short nitriding time and might also be influenced by the lower deformation energy storage resulting from the warm-rolling of high-silicon steel strips.

Furthermore, through comparative analysis of the variations in nitriding time and texture, it is observed from Figure 6a–f that there was no discernible pattern of change in the microstructure and texture as the nitriding time increased. The prevailing texture was dominated by γ and {113}<361>. Notably, the {111}<112> texture component, which plays a favorable role in subsequent abnormal growth of Goss grains, accounts for approximately 12%, lower than the 20% typically observed in traditional 3% Si nitrided steel. This indicates that the warm rolling system employed for high-silicon steel attenuates the {111}<112> texture component. The nitriding time has minimal effect on both grain size and texture, mainly affecting the nitrogen content within the sample. 

### 3.2. Precipitation of Second-Phase Particles

The presence of second-phase particles in oriented silicon steel influences the development of strong Goss textures during secondary recrystallization. From the perspective of the metallurgical composition system, the intrinsic inhibitor for Cu-containing steel, should include Cu and S precipitates. Figure 7 illustrates the precipitated particles within the hot-rolled plate. The findings indicate that the majority of particles present in the hot-rolled sheet consisted of Cu and S precipitates, with an average size of approximately 50 nm. Due to the adoption of a slab heating process at 1150 °C before hot rolling, the lower heating temperature caused a reduction in the amount of Cu and S atoms dissolved in the high silicon. Consequently, it greatly impaired the nucleation force during the subsequent cooling process, resulting in fewer precipitated particles.

Figure 8a–c presents the morphology of the precipitated particles observed at different nitriding times. The figure illustrates that, as the nitriding time increased, a greater number of nitrogen atoms entered the sample and reached deeper regions. After nitriding for 60 s, a quantity of second-phase particles precipitated on the surface of the test steel, as depicted in Figure 8a. With nitriding durations of 90 s and 120 s, the second-phase particles started to cluster together. The majority of these precipitated second-phase particles exhibited a regular square shape and had a small size ranging from 20 to 50 nm, as evidenced in Figure 8b,c. Meanwhile, EDS spectrum analysis inferred that these small square particles consisted of Si_3_N_4_ and (Al,Si)N. However, Si_3_N_4_ has a tendency to change into (Al,Si)N under specific conditions due to its unstable nature. Moreover, Si_3_N_4_ particles formed after nitriding are not evenly distributed throughout the thickness direction of the sample; instead, they predominantly concentrate in the surface layer. The conversion of Si_3_N_4_ into (Al,Si)N helps achieve a more uniform distribution of N atoms, and the resulting dispersed fine (Al,Si)N particles serve as inhibitors. This transformation occurs within the temperature range of 700 °C–750 °C [19]. Since the nitriding temperature selected for this experiment was 750 °C, some Si_3_N_4_ particles have already undergone conversion to (Al,Si)N. However, a complete conversion can only be accomplished during high-temperature annealing processes.

Table 1 presents the nitrogen content of samples with different nitriding times, reflecting variations in the concentration of inhibitors. The nitrogen content fell within the range of 130 ppm to 246 ppm. The recommended N distribution in conventional nitrided steel typically ranges from 130 ppm to 240 ppm [20,21]. While the nitrogen content after nitriding for 60 s and 90 s fell within this range, the morphology of precipitates after nitriding for 60 s indicates that the particle density of precipitates was notably lower compared to that of conventional nitrided steel.

### 3.3. Microstructure and Magnetic Properties of the Final Annealed Plate

Figure 9a–d displays the final macrostructure of the annealed sheet. It is obvious from the figure that the samples not subjected to nitriding had an average grain size in the range of 2–3 mm, with the texture types basically identical to those observed after primary recrystallization, as shown in Figure 9a. Based on these observations, it can be deduced that secondary recrystallization did not occur in the non-nitrided samples. In contrast, all samples subjected to nitriding exhibited varying degrees of secondary recrystallization. The sample exposed to 60 s of nitriding displayed a secondary recrystallization ratio exceeding 80%, with an average secondary grain size of 8 mm. However, fine grains were still discernible in specific localized regions of the sample, indicated by the green color in Figure 9b. The sample treated with 90 s of nitriding exhibited the most advanced secondary recrystallized structure and the most distinct Goss textures. Moreover, the macroscopic images clearly reveal that all grains in the sample had undergone secondary recrystallization, with an average grain size ranging between 9 mm and 11 mm, as shown in Figure 9c. The secondary grains accounted for a lower proportion in the sample subjected to 120 s of nitriding. Moreover, a significant number of fine grains existed in the finished sheet, as shown by the blue arrow in Figure 9d.

The magnetic properties at 50 Hz of the finished sheet made from grain-oriented high-silicon steel is presented in Table 2. The magnetic properties of the B_8_, after nitriding, fell within the range of 1.462 T and 1.625 T, while the magnetic properties of the B_50_ approach the saturated magnetic strength Bs (1.80 T) of the Fe-6.5% Si alloy. Thus, it can be concluded that the samples subjected to 90 s of nitriding exhibited complete secondary recrystallization and the best magnetic properties. 

### 3.4. Microstructure and Texture Evolution of Samples Extracted by Interrupted Annealing

Figure 10a–e shows the EBSD orientation maps that underwent 90 s of nitriding during an interrupted high-temperature annealing. It can be seen from Figure 10a,b that no abnormal growth occurred during annealing at 850 °C or 900 °C. In these temperature conditions, the sample was dominated by γ and {113}<361> grains, with a small number of Goss orientation grains interspersed within the γ grains. At an annealing temperature of 950 °C, certain Goss grains situated in the upper and lower surface layers displayed a significantly accelerated growth rate compared to the neighboring grains, indicating the possibility of secondary recrystallization. At this juncture, the dimensions of these Goss grains exceeded 200 µm, as illustrated in Figure 10c. As the annealing temperature continued to rise, the abnormally grown Goss grains progressively extended beyond the thickness of the sheet, leading to the merging of adjacent smaller grains. Figure 10d illustrates that secondary recrystallization had already occurred. Upon reaching the annealing temperature of 1100 °C, secondary recrystallization was completed, forming a secondary recrystallized structure dominated by a Goss orientation, as shown in Figure 10e.

Figure 11a,b illustrates the misorientation distribution between Goss grains and neighboring grains during the interrupted annealing process at 900 °C and 950 °C, while Figure 12 displays the misorientation distribution at 1000 °C. It is evident that the frequency of grain boundary misorientations between Goss and adjacent grains falling within the 20° to 45° range was significantly higher compared to random grains. This situation creates favorable conditions for the subsequent abnormal growth of Goss grains. In comparison to the interrupted annealing at 900 °C, the sample annealed at 950 °C exhibited a greater number of grain boundaries falling within the 20° to 45° range, as depicted in Figure 11a,b. During the initial stages of secondary recrystallization, particularly at annealing temperatures of 1000 °C, the proportion of Goss oriented grains relative to the surrounding grain boundaries within the 20° to 45° range was notably high at 62.7%. This observation indicates the presence of more “free” grain boundaries surrounding the Goss grains, as demonstrated in Figure 12. 

Furthermore, the average grain size and percentage of {111} plane textures in the four different types of samples following interrupted annealing are shown in Figure 13. The variations in average grain sizes reveal that the primary recrystallization occurred during annealing in the range of 900 °C–1000 °C, resulting in a slight increase in the average grain size for all four types of samples. During the subsequent secondary recrystallization phase within the 1000 °C to 1100 °C range, the average grain size in nitrided samples increased drastically, whereas the increase in samples without nitriding occurred at a much slower rate, corresponding to normal grain growth. When comparing the three types of nitriding samples, it can be seen that during the secondary recrystallization stage, the samples nitrided for 90 s showed the highest grain growth rate and the largest average grain size. Conversely, the sample nitrided for 120 s experienced a delayed onset of secondary recrystallization temperature due to the strong pinning ability of the inhibitor, consequently leading to a slower grain growth rate. The samples nitrided for 60 s exhibited a grain growth rate that falls between the other two sample types, as shown in Figure 13a. The secondary recrystallization in oriented silicon steel is achieved through the merging of Goss grains with surrounding γ grains; thus, quantitative statistics on the percentage of {111} plane textures can also provide insights into why complete secondary recrystallization does not take place in the three sample types. The data analysis on the percentage of {111} plane textures suggests an upward trend in the percentage content of {111} surface texture without nitriding as the annealing temperature rises. Nonetheless, Goss-oriented grains did not exhibit a growth advantage, suggesting that achieving secondary recrystallization solely based on inherent inhibitors is challenging for the samples without nitriding. After annealing at 1000 °C, there was a declining trend in the percentage content of {111} surface texture in the three categories of nitrided samples. Therefore, it can be inferred that the inclusion of inhibitors played a pivotal role in stimulating the abnormal growth of Goss grains. The distinct rates of decline (indicated by straight slopes) in the percentage of {111} plane textures among the three sample types signify differing growth rates of the Goss-oriented grains. These growth rates are associated with the varying capacity of the inhibitor to impede the surrounding boundaries, which is influenced by the quantity of the inhibitor present. In the samples subjected to 90 s of nitriding, the percentage of {111} plane textures exhibited a nearly linear decline, showing a better correlation between the number of inhibitors and the abnormal growth of Goss grains. This is beneficial for achieving complete secondary recrystallization of Goss grains. On the other hand, the decreasing trend of the {111} texture percentage content in the sample nitrided for 60 s was less pronounced compared to sample nitride for 90 s, indicating an insufficient addition of inhibitors. Meanwhile, in the samples subjected to 120 s of nitriding, the percentage of {111} textures showed a minimal difference between 1000 °C and 1050 °C, exhibiting only a slight decrease. Hence, it is inferred that the sample nitrided for 120 s experienced a delay in the process of secondary recrystallization, which is attributed to the large number of inhibitors leading to excessive pinning at the grain boundary.

## 4. Discussion

Achieving accurate Goss-oriented grains during high-temperature annealing is a crucial research objective as it is the key characteristic of grain-oriented silicon steel, enabling the formation of a sharp {110}<001> texture through secondary recrystallization and ensuring favorable magnetic properties. The secondary recrystallization behavior of Goss textures in grain-oriented silicon steel has been a subject of debate for many years, leading to the proposal of two prominent theories: the high-energy (HE) grain boundary theory and the coincident site lattice (CSL) theory [22,23,24,25,26]. Both theories believe that special grain boundaries with enhanced mobility are the main drivers for the anomalous growth of Goss grains. However, while the HE grain boundary theory emphasizes the influence of grain boundaries with orientation differences between 20° and 45° on the Goss secondary recrystallization, the CSL theory suggests that Goss grains exhibiting CSL grain boundary characteristics tend to possess fewer solute atoms clustered at the grain boundaries, resulting in weaker pinning forces. Therefore, these specific grain boundaries preferentially detach pinning, causing abnormal growth of Goss grains. Notably, Σ9 grain boundaries with an orientation of 35°<110> exhibit enhanced mobility. Nevertheless, the presence of Σ9 grain boundaries was minimal, constituting less than 5% of the total boundaries in this experiment. This suggests that the influence of CSL boundaries on the abnormal growth behavior of Goss grains is limited. Conversely, HE grain boundaries emerge as a pivotal factor in facilitating the abnormal growth behavior of Goss grains. The frequency of distribution of Goss-oriented grains at HE grain boundaries in the range of 20°–45° was notably higher compared to that at random grain boundaries. This observation indicates that the preferential growth behavior of Goss-oriented grains is achieved by the high mobility provided by HE grain boundaries.

Considering the primary recrystallization structure, texture, and the impact of inhibitors on secondary recrystallization behavior, it becomes apparent that the primary recrystallization stage governs the size of the secondary recrystallization seeds and grains. On the other hand, the inhibitors play a crucial role in determining both the inhibitory and driving forces of secondary recrystallization. Despite the fact that the strength of the γ texture in grain-oriented high-silicon steel is weaker compared to that of traditional 3 wt% Si grain-oriented silicon steel, there is a higher distribution frequency of Goss-oriented grains at HE grain boundaries within the range of 20°–45°. In addition, the secondary recrystallization temperature exceeds 1000 °C, providing Goss-oriented grains with a favorable orientation environment and thermodynamic temperature environment, ultimately forming a secondary recrystallization structure with a sharp Goss texture. During the secondary recrystallization process of Goss-oriented grains, a preferential growth mechanism comes into play, where the high mobility of HE grain boundaries within the range of 20° and 45° enables the preferential growth of Goss-oriented grains.

## 5. Conclusions

This study comprehensively investigated the evolution of microstructure, texture, and the magnetic properties of grain-oriented 6.5% Si electrical steels formed by rolling and incorporation of intrinsic inhibitors and additional inhibitors. This research highlights the crucial role of nitriding quantity in achieving complete secondary recrystallization. In instances where additional nitrogen is absent, the intrinsic inhibitors alone do not lead to secondary recrystallization. However, when the nitriding duration is 90 s and the nitriding amount is 185 ppm, a complete secondary recrystallization structure with a strong Goss texture enables the finished products have excellent magnetic properties. The preferential growth behavior of Goss-oriented grains primarily depends on the high mobility of HE grain boundaries. With the increase in annealing temperature, there is a gradual increase in the occurrence of HE grain boundaries within the range of 20°–45° that are associated with Goss grains. Furthermore, during the secondary recrystallization process at a temperature of 1000 °C, there is a substantial occurrence of HE grain boundaries within the 20°–45° range, accounting for 62.7%. This prevalence creates favorable conditions for the abnormal growth of Goss grains, ultimately leading to the formation of a secondary recrystallization structure dominated by a strong Goss texture. The present study’s findings provide a novel and efficient way to optimize the recrystallization texture and improve the magnetic properties of 6.5% Si grain-oriented electrical steels.

## Figures and Tables

**Figure 1 materials-16-06731-f001:**
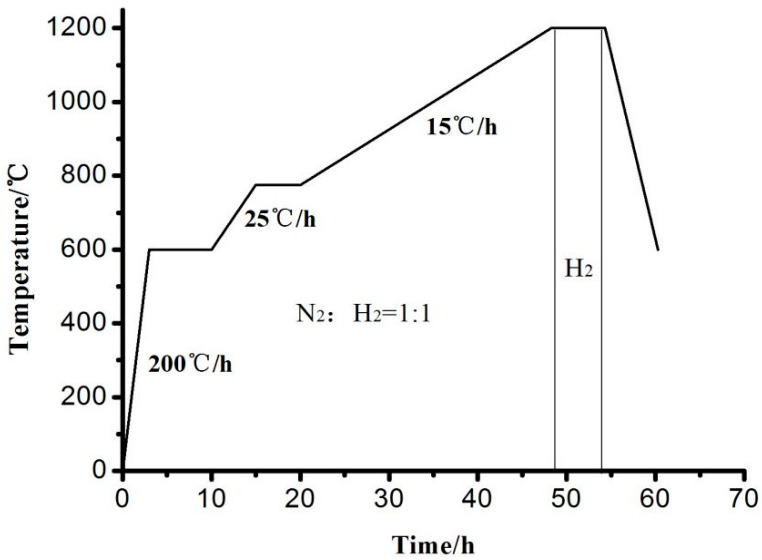
High-temperature annealing process.

**Figure 2 materials-16-06731-f002:**
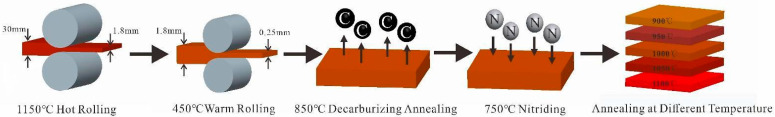
The schematic diagram of preparation process.

**Figure 3 materials-16-06731-f003:**
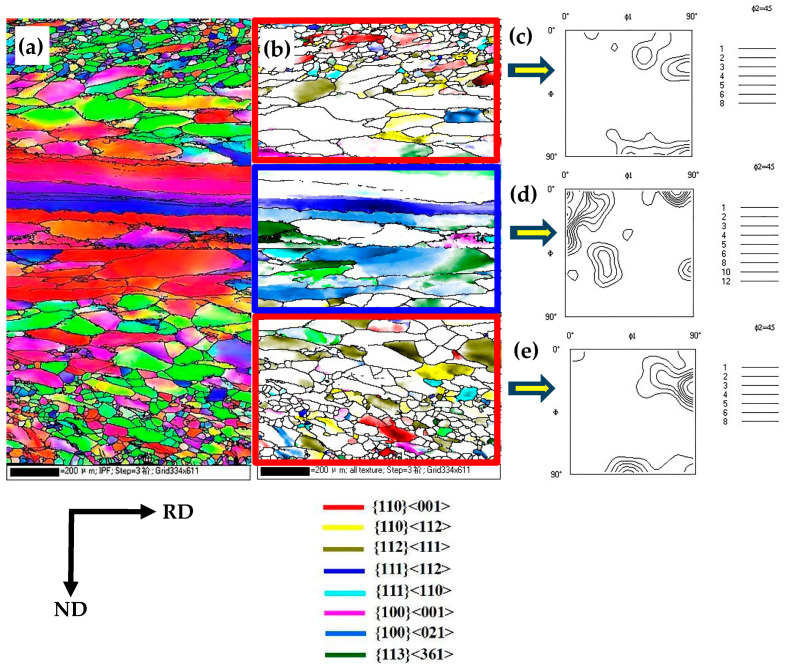
EBSD orientation maps of hot-rolled sheets in the lateral section. (**a**) EBSD IPF map; (**b**) several texture components of hot-rolled sheets; (**c**) the texture (φ_2_ = 45° section of ODFs) in upper surface layer of hot-rolled sheets; (**d**) the texture (φ_2_ = 45° section of ODFs) in center layer of hot-rolled sheets; (**e**) the texture (φ_2_ = 45° section of ODFs) in lower surface layer of hot-rolled sheets.

**Figure 4 materials-16-06731-f004:**
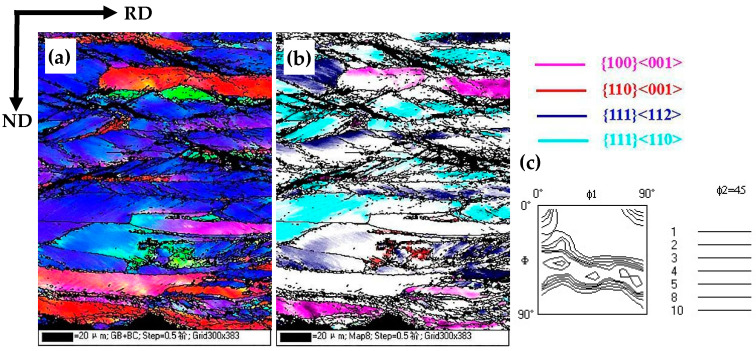
EBSD orientation maps of cold-rolled sheets in the lateral section: (**a**) EBSD IPF map; (**b**) several texture components colored in orientation maps of cold-rolled sheet; (**c**) the texture (φ_2_ = 45° section of ODFs) of cold-rolled sheet.

**Figure 5 materials-16-06731-f005:**
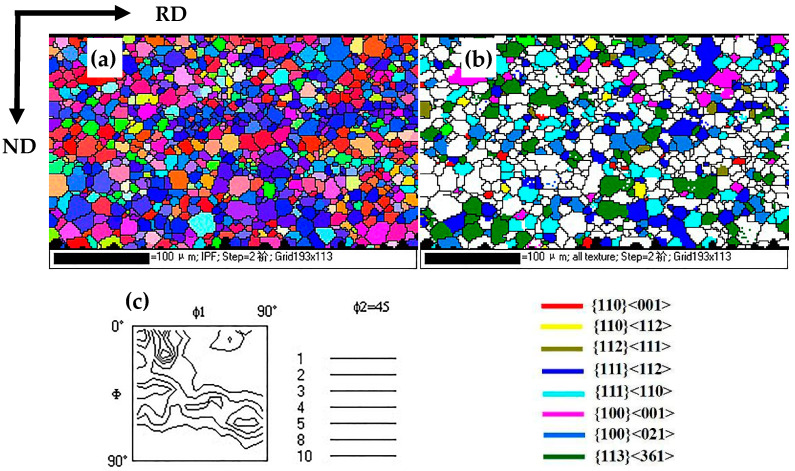
EBSD orientation maps of decarburized sample. (**a**) EBSD IPF map; (**b**) several texture components of decarburized sample; (**c**) the texture (φ_2_ = 45° section of ODFs) of decarburized sample.

**Figure 6 materials-16-06731-f006:**
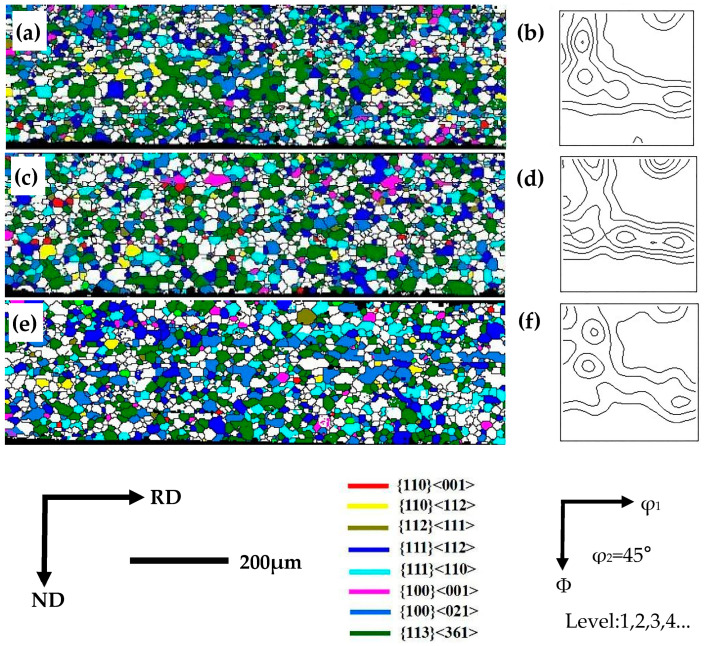
EBSD orientation maps at different nitriding times. (**a**) EBSD orientation maps after nitriding for 60 s; (**b**) the texture (φ_2_ = 45° section of ODFs) after nitriding for 60 s; (**c**) EBSD orientation maps after nitriding for 90 s; (**d**) the texture (φ_2_ = 45° section of ODFs) after nitriding for 90 s; (**e**) EBSD orientation maps after nitriding for 120 s (**f**) the texture (φ_2_ = 45° section of ODFs) after nitriding for 120 s.

**Figure 7 materials-16-06731-f007:**
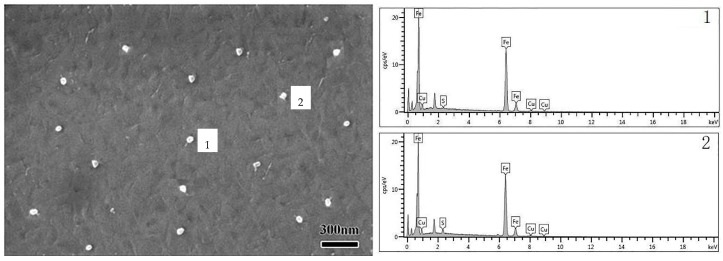
Precipitated particles in hot-rolled plate.

**Figure 8 materials-16-06731-f008:**
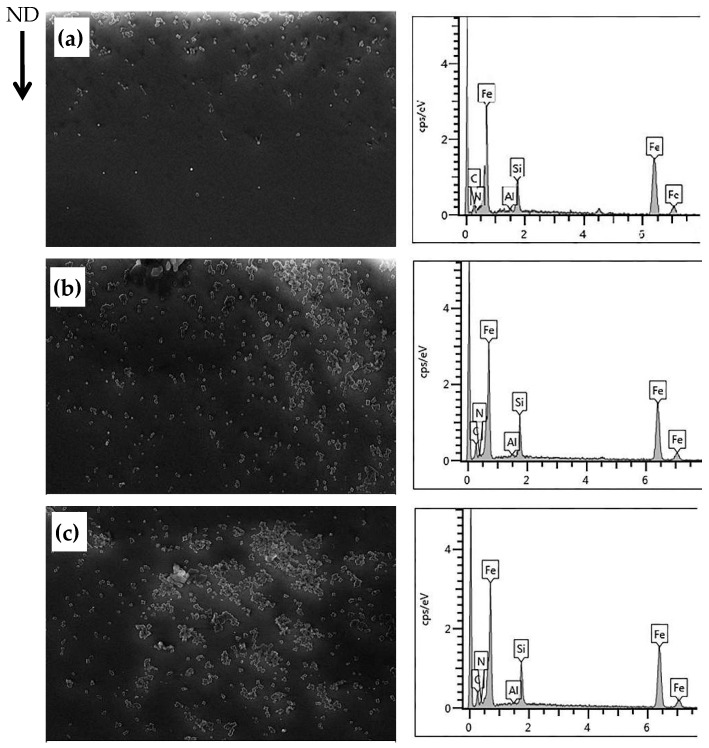
Particle morphology at different nitriding times. (**a**) Nitriding for 60 s; (**b**) nitriding for 90 s; (**c**) nitriding for 120 s.

**Figure 9 materials-16-06731-f009:**
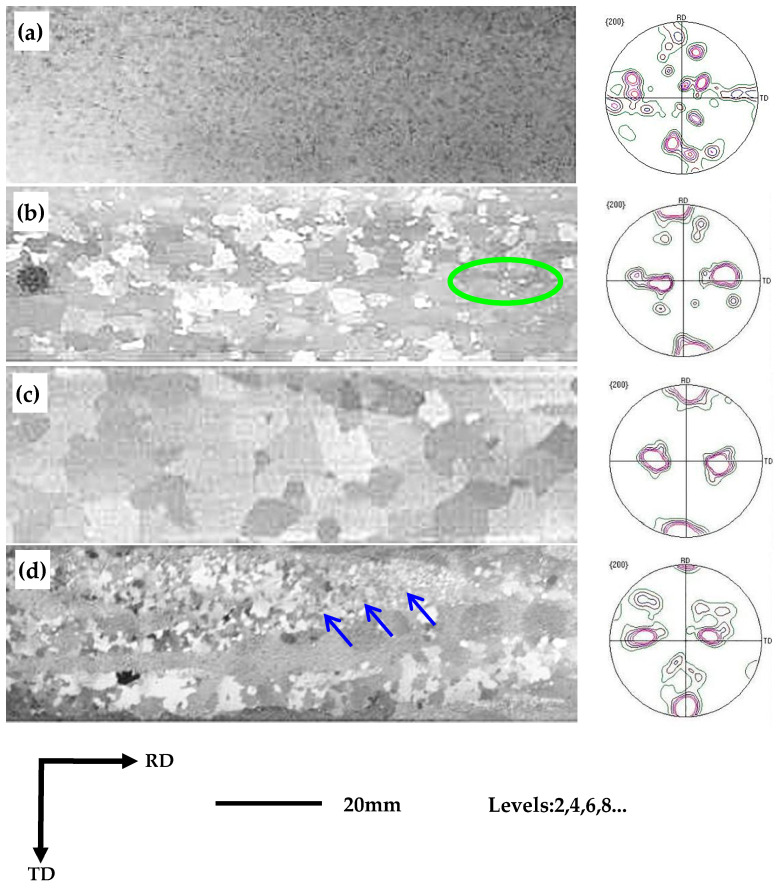
Macrostructure and texture of the final annealed sheet. (**a**) Nitriding for 0 s; (**b**) nitriding for 60 s; (**c**) nitriding for 90 s; (**d**) nitriding for 120 s.

**Figure 10 materials-16-06731-f010:**
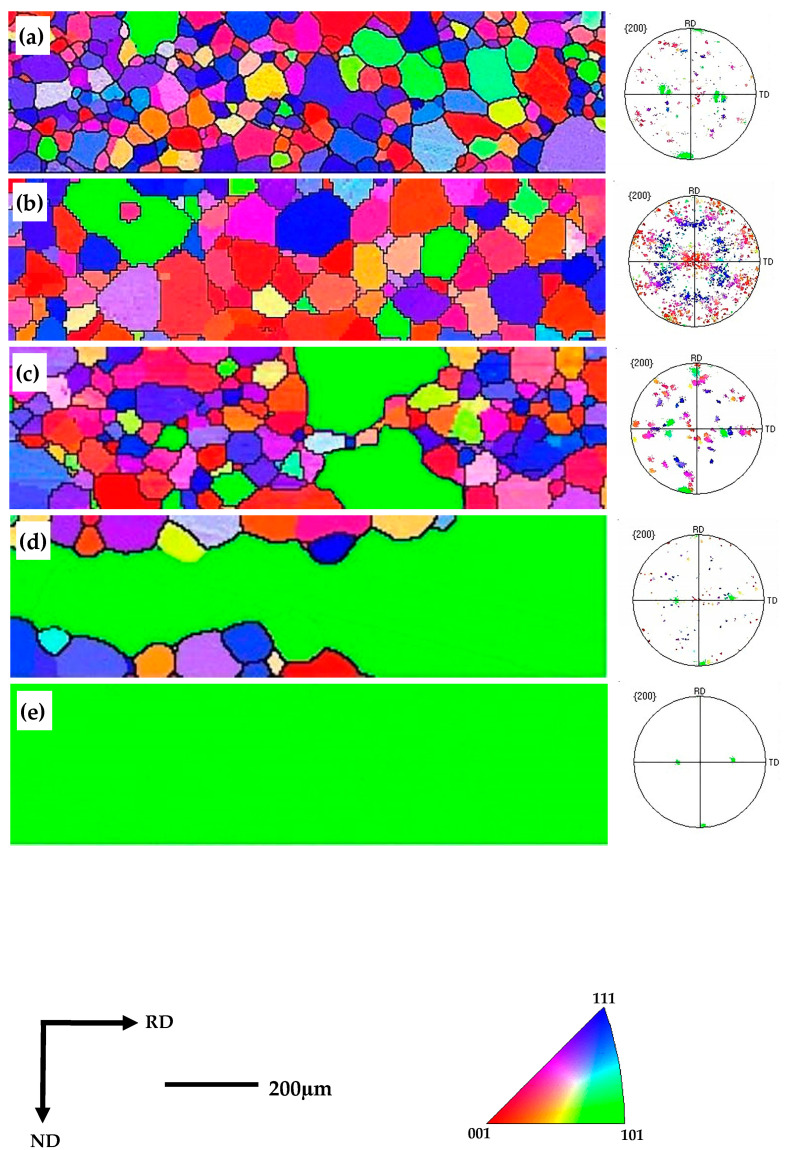
EBSD orientation maps of samples after nitriding for 90 s extracted by interrupted annealing. (**a**) Annealing at 900 °C; (**b**) annealing at 950 °C; (**c**) annealing at 1000 °C; (**d**) annealing at 1050 °C; (**e**) annealing at 1100 °C.

**Figure 11 materials-16-06731-f011:**
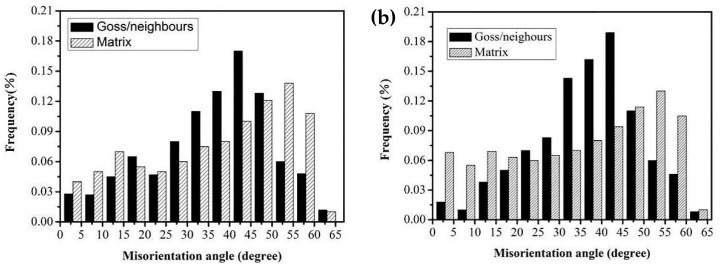
Misorientation distribution. (**a**) Annealed at 900 °C; (**b**) annealed at 950 °C.

**Figure 12 materials-16-06731-f012:**
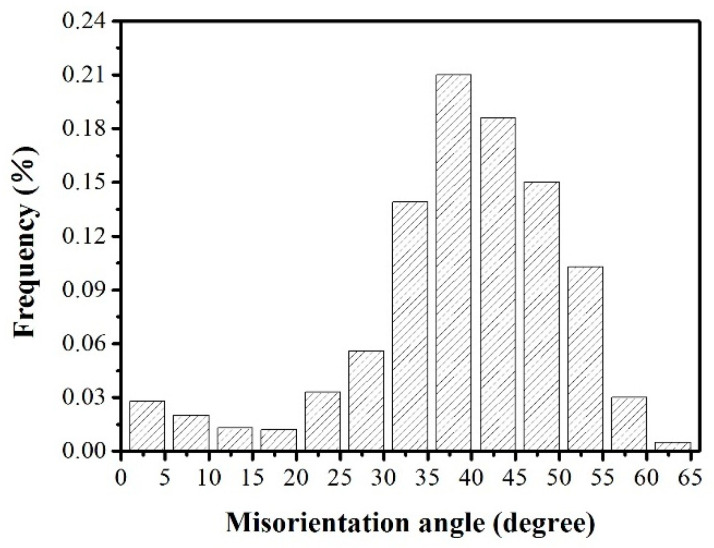
Misorientation distribution of recrystallized grains annealed at 1000 °C.

**Figure 13 materials-16-06731-f013:**
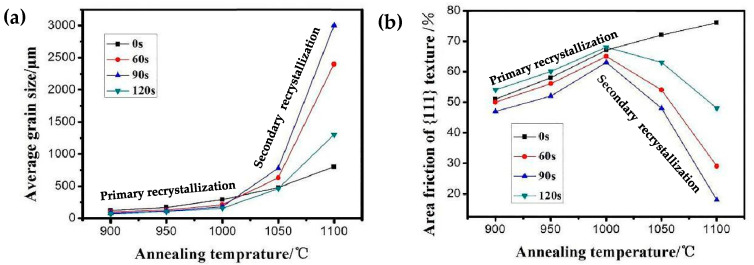
Statistics of average grain size and {111} texture percentage at different annealing times. (**a**) Average grain size at different annealing times; (**b**) 111} texture percentage at different annealing times.

**Table 1 materials-16-06731-t001:** Statistics of nitriding amount and average grain size at different nitriding times.

Nitriding time/s	60	90	120
Average grain size/μm	17.8	21.1	21.3
Nitriding amount/ppm	130	185	246

**Table 2 materials-16-06731-t002:** Magnetic properties of the final annealed sheet.

Nitriding Time (s)	B_8_ (T)	B_50_ (T)	P_1.0/50_ (W/kg)	P_1.5/50_ (W/kg)
0	1.462	1.619	0.892	1.782
60	1.538	1.722	0.587	1.473
90	1.625	1.789	0.381	1.279
120	1.504	1.691	0.775	1.618

## Data Availability

The data is available from the corresponding author upon reasonable request.

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
