# Peer review of "Evolution of Microstructure and Texture in Grain-Oriented 6.5% Si Steel Processed by Rolling with Intrinsic Inhibitors and Additional Inhibitors"

_materials, 2023, doi:10.3390/ma16206731_

Round 1

Reviewer 1 Report

The manuscript reported on the characterization of Si electrical steel dominated by Goos texture. Interesting results were presented, however the following issues exist with the manuscript. A general proofreading is required to improve the occasional errors, some of which were indicated in the attached pdf. Evidence for one of the inhibitors, CuS, were not presented. Cu and S are evident in the elemental analysis, but further characterization is required to prove the formation of CuS. The introduction focused exclusively on manufacturing of Si steel, however its electrical performance important for electrical steel is currently missing. Detailed comments are attached and should be thoroughly addressed for further consideration of the manuscript.

Minor editing of English language required

Author Response

Dear reviewer and editor:

Thank you very much for your comments on our manuscript. The questions are very helpful for improving our paper, We have studied the comments carefully and have made extensive modification which we hope meet with approval. In addition, the English has also been polished by professional institutions. The revised contents in the text corresponding to the comments proposed by the editor and reviewer are marked in red. The comments (marked in blue) and responses (marked in black) are as following: 

Reviewer 1: 
Comment (C) 1: The title is self-explanatory but is too long and repetitive. One word adequately explains the "inherent and addtional inhibitors". The Si steel was "rolling-processed", therefore the word "method" can be removed.
Response (R) 1: Adopting your suggestion, the word "method" is removed.

Comment (C) 2: What other parameters should be considered during steel manufacturing? What are optimal concentrations of Fe and Si? These details require a clear introduction in the manuscript.

Response (R) 2: The preparation process of grain-oriented silicon steel is complex and has many influencing factors. The grain size and texture after primary recrystallization, rolling process, nitriding process, and high-temperature annealing process, which all have significant impact on the development of Goss texture. The preparation of grain-oriented high silicon steel is more difficult due to its high silicon content, low deformation energy storage, large primary recrystallization grain size and insufficient pinning force, therefore, there is very little research on grain-oriented high silicon steel. On the one hand, there is no mature experience in the preparation process of grain-oriented high silicon steel, and the effect of process parameters on the development of Goss texture and magnetic properties of grain-oriented high silicon steel is still under research. On the other hand, if the introduction is written from the perspective of factors affecting the magnetic properties of electrical steel, the length of the introduction will be relatively large and the focus will not be prominent.

In theory, when the silicon content increases to 6.5%, the electrical steel has excellent soft magnetic properties. As the electrical steel ratio between Fe and Si is relatively popular and scientific, therefore, there is no explanation in the paper. Partially adopting your suggestions, with some adjustments made in the introduction section.

Comment (C) 3: What is novel in all these research efforts? Moreover a brief introduction of the Goss texture grain is missing to educate a non expert readership.

Response (R) 3: This paragraph aims to explain that due to the poor plasticity of high silicon steel at room temperature, warm rolling technology needs to be introduced. Warm rolling results in lower deformation energy storage, coarse primary recrystallization grain size, and difficulty in preparing grain-oriented high silicon steel using inhibitors. Therefore, most scholars have turned to studying non-oriented high silicon steel, and there is not much innovation in the preparation of oriented high silicon steel.

In addition, adopting your suggestion, the description of Goss texture has been introduced in the paper.

Comment (C) 4: Si concentration appears without prior introduction of optimal concentrations. Why the majority of reports utilized a 6.5% concentration for best steel performance and why not use a lower and hence more economic concentration of Si?

Response (R) 4: Theoretically, 6.5% Si has the best soft magnetic performance, so the silicon content in this study is set to 6.5%. The Si concentration in this article is theoretically optimal.

Comment (C) 5: The introductory provided several reports on the manufacturing of Si steel. However, the grain orientation is important for electrical steel and reports on the characterization of its electrical performance are missing from the manuscript. There have been several reports, see that provide electrical activity evidence as well as results to improve upon and introduce this important aspect of electrical Si.

Response (R) 5: The research of this article is the preparation of grain-oriented high silicon steel, with a focus on the changes in microstructure and texture during the preparation process. The focus of this article differs from the several literature provided. For grain-oriented silicon steel, the main development is Goss texture. The Goss texture is a beneficial texture, as explained in the paper. Considering that the effect of Goss texture on magnetic properties of grain-oriented silicon steel is relatively scientific, there is not much description.

   Adopting your suggestion, the effect of Goss texture on magnetic properties has been introduced in the paper.

Comment (C) 6: It would improve the readability of the manuscript if each figure is labeled with the nitriding time.

Response (R) 6: Adopting your suggestion, each figure is labeled with the nitriding time.

Comment (C) 7: Cu and S are evident in the elemental mapping, however more evidence of CuS are required. Unless further e.g. XRD or XPS characterization is presented demonstrating the presence of CuS, the paragraph should be rephrased.

Response (R) 7: Adopting your suggestion, the paragraph is rephrased. 

Comment (C) 8: Evidence for formation of CuS were not presented. Please see previous comments and rephrase accordingly.

Response (R) 8: Adopting your suggestion, the corresponding paragraph has been rephrased.

Reviewer 2 Report

In this paper, the authors present a study of the microstructure in grain-oriented silicon steel. This study deserves special attention because it emphasizes the role of the amount of nitriding to achieve complete secondary recrystallization.

However, in the course of reading the work, I had some questions:

1.                      Figure 6 shows the EBSD of three samples obtained at different nitriding times. It can be seen from the presented data that the nitriding time does not affect the change in grain size. However, I would like to see a comparison of the data obtained with the material processed without nitriding.

2.                      At 60 seconds, the grain size is 17.8 mm, and at 90 seconds, 21.1 mm. Maybe with a greater increase in the nitriding time, the difference in grain size would be more significant? What results can be obtained with three-minute nitriding?

3.                      Why do the authors compare separately formed particles with the Si3N4 compound by means of elemental analysis? Nitrogen is present in the EDS diagrams, but is it allowed that it can be nitrogen diffused into the metal or another compound?

4.                      At what rate did the sample cool down after heat treatment? And whether the cooling rate can affect the properties of silicon steel.

5.                      The authors report that each stage of the heat treatment of steel was carried out during a specific time. For example, the temperature of 600 degrees was maintained for 4 hours and 86 minutes. Were these values obtained from multiple studies or taken from other studies? How much the structure of silicon steel can change when the processing time changes.

6.                      What is the reason for the change in the magnetic properties of steel with an increase in the level of nitriding during processing?

7.                      I think that the authors need to expand the list of references with the latest relevant research.

Author Response

Dear reviewer and editor:

Thank you very much for your comments on our manuscript. The questions are very helpful for improving our paper, We have studied the comments carefully and have made extensive modification which we hope meet with approval. In addition, the English has also been polished by professional institutions. The revised contents in the text corresponding to the comments proposed by the editor and reviewer are marked in red. The comments (marked in blue) and responses (marked in black) are as following: 

Reviewer 2:

Comment (C) 1: Figure 6 shows the EBSD of three samples obtained at different nitriding times. It canbe seen from the presented data that the nitriding time does not affect the change in grain size. However, I would like to see a comparison of the data obtained with the material processed without nitriding.
Response (R) 1: The samples without nitriding have little difference in grain size and texture type, as shown in the following figure. The grains are slightly smaller, but the texture are still consisted of γ-fiber and {100}<021>~{113}<361>. Considering that figure 6 and table 1 need to correspond to each other, therefore, this image is not attached.

Comment (C) 2: At 60 seconds, the grain size is 17.8 mm, and at 90 seconds, 21.1 mm. Maybe with agreater increase in the nitriding time, the difference in grain size would be more significant? What results can be obtained with three-minute nitriding?

Response (R) 2: The previous experimental results indicate that the nitriding time is too long, the nitriding amount is too large, delaying the secondary recrystallization of Goss, resulting in Goss not being able to fully complete the secondary recrystallization. In addtion, if the nitriding time is too long, the grain size of primary recrystallization will increase, which is also adverse to the  secondary recrystallization of Goss grains. Usually, the nitriding time of 3% Si oriented silicon steel ranges from 60s to 120s. This article draws inspiration from the nitriding time of 3% Si oriented silicon steel and previous experimental research, therefore the selected nitriding time is 60s to 120s.

Comment (C) 3: Why do the authors compare separately formed partidles with the Si3N4 compoundby means of elemental analysis? Nitrogen is present in the EDS diagrams, but is it allowed that it can be nitrogen diffused into the metal or another compound?

Response (R) 3: This is a very professional technical issue. The statement of Si3N4 transformed into (Al,Si)N is based on previous research work, quotes the research conclusions and experiences of reference 17. The distribution of inhibitors plays an important role in the pinning of grain boundaries and the development of Goss in oriented silicon steel. The transformation and decomposition of the Si3N4 particle has a significant impact on the distribution of inhibitors. This paragraph is cited to illustrate that the nitriding temperature selected for this experiment is appropriate, and the distribution of inhibitors is conducive to the abnormal growth of Goss grains in the future.

[17] Ushigami. Y, Kurosawa. F, Komatsu. H. Process for preparation of oriented electrical steel sheet having high flux density: US Patent, 5888314[P]. 1999.

Comment (C) 4: At what rate did the sample cool down after heat treatment? And whether the cooling rate can affect the properties of silicon steel.

Response (R) 4: Cooling method adopts furnace cooling. Due to the complete secondary recrystallization of the test steel after high-temperature annealing, the cooling rate will not affect the microstructure and texture.

Comment (C) 5: The authors report that each stage of the heat treatment of steel was carried out during a specific time. For example, the temperature of 600 degrees was maintained for 4 hoursand. Were these values obtained from multiple studies or taken from other studies? How much the structure of silicon steel can change when the processing time changes.

Response (R) 5: The preparation process of oriented silicon steel is long and difficult, requiring strict control of process parameters such as inhibitors, primary recrystallization grain size and texture, annealing atmosphere, and heating rate. When the processing time changes, secondary recrystallization is difficult to complete, and the magnetic properties of the test steel plate will be greatly affected. The selection of heat treatment parameters for experimental steel is obtained through extensive experimental research, at the same time, it also draws on the traditional experience of preparing 3% Si oriented silicon steel

Comment (C) 6: What is the reason for the change in the magnetic properties of steel with an increase in the level of nitriding during processing?

Response (R) 6: With an increase in the level of nitriding during processing, the enhanced ability of inhibitors to pin grain boundaries provides favorable conditions for secondary recrystallization of Goss oriented grains. Beneficial texture enhancement leads to an increase in the magnetic properties of the strip steel.

Comment (C) 7: I think that the authors need to expand the list of references with the latest relevant research.

Response (R) 7: Adopting your suggestion, the latest relevant references have been added in the corresponding paragraphs.

By the way, thank you very much for the very careful revision of the first manuscript.

Yours sincerely,

Ruiyang Liang

School of Materials Science and Engineering,

Liaoning University of Technology

2023-09-28

Round 2

Reviewer 1 Report

The authors provided line-by-line revisions to the concerns of the review. Many of the concerns have been addressed and the manuscript has been improved. In addition to the presented results and analysis of grain size and orientation of Si steels, the manuscript will benefit from a broader reference to the application of electrical Si steels supported by demonstrated results as suggested by relevant literature. In this way, the presented results can impart a greater impact for further research and development of Si electrical steels.

Author Response

Dear reviewer and editor:

Thank you very much for your comments on our manuscript. Adopting your suggestion, the latest relevant references have been added in the paper. The revised contents in the text corresponding to the comments proposed by the editor and reviewer are marked in red. In addition, this paper has been carefully edited by a native English-speaking editor at Sagesci. The CERTIFICATE OF ENGLISH EDITING can be found in the appendix.

Reviewer 2 Report

This version of manuscript significantly improved, and I can recommend it to publication

Author Response

(The authors gave the same response as above.)
